# UPAR: A Kantian-Inspired Prompting Framework for Enhancing Large Language Model Capabilities

## Abstract

Large Language Models (LLMs) have demonstrated impressive inferential capabilities, with numerous research endeavors devoted to enhancing this capacity through prompting. Despite these efforts, a unified epistemological foundation is still conspicuously absent. Drawing inspiration from Kant's a priori philosophy, we propose the UPAR prompting framework, designed to emulate the structure of human cognition within LLMs. The UPAR framework is delineated into four phases: "Understand", "Plan", "Act", and "Reflect", enabling the extraction of structured information from complex contexts, prior planning of solutions, execution according to plan, and self-reflection. This structure significantly augments the explainability and accuracy of LLM inference, producing a human-understandable and inspectable inferential trajectory. Furthermore, our work offers an epistemological foundation for existing prompting techniques, allowing for a possible systematic integration of these methods. With GPT-4, our approach elevates the accuracy from COT baseline of 22.92% to 58.33% in a challenging subset of GSM8K, and from 67.91% to 75.40% in the causal judgment task. Without using few-shot examples or external tools, UPAR significantly outperforms existing prompting methods on SCIBENCH, a challenging dataset containing collegiate-level mathematics, chemistry, and physics scientific problems.

## 1 INTRODUCTION

Large Language Models (LLMs) have showcased capabilities parallel to human performance across diverse tasks encompassing programming, academic evaluations, common sense reasoning, and professional undertakings (OpenAI, 2023; Bubeck et al., 2023; Zhao et al., 2023b). The emergent in-context learning ability (Brown et al., 2020; Wei et al., 2022) of LLMs has catalyzed the evolution of prompt engineering, aiming to further amplify their potential (Liu et al., 2021; Dong et al., 2022; Qiao et al., 2023; Dong et al., 2023; Mialon et al., 2023). Despite this advancement, a significant portion of current research on prompting remains largely heuristic and intuitive, frequently devoid of a profound epistemological foundation. To address this concern, this paper introduces the UPAR framework, inspired by the transcendental philosophy of Immanuel Kant. The proposed framework aspires not only to enhance the reasoning competencies of LLMs but also to offer a consolidated, epistemology-based structure for existing prompt techniques.

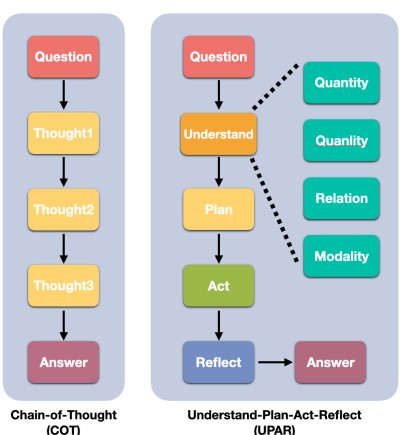

Figure 1: Comparison of UPAR to Chain-of-Thought. UPAR includes four parts: Understand, Plan, Act, and Reflect, allowing LLMs to simulate the mental structure of human.

Prompt Engineering emerges as an efficacious approach to augment the performance of LLMs without the necessity for model retraining. This enhancement is achieved by the construction of specialized instructive prompts, such as chain-of-thought (Wei et al., 2023), self-consistency (Wang et al., 2023b), and knowledge generation (Liu et al., 2022), among others. These strategies hold the potential to mitigate various shortcomings inherent in contemporary large language models, such as inexplicability (Zhao et al., 2023a), hallucinations (Bang et al., 2023; Liang et al., 2022; McKenna et al., 2023; Li et al., 2023; Ji et al., 2023; Rawte et al., 2023), complex causal reasoning (Hobbhahn et al., 2022; Willig et al., 2022; Zečević et al., 2023; Kıcıman et al., 2023), and limited mathematical ability (Lu et al., 2023b; Dziri et al., 2023). Nevertheless, the majority of existing prompt methodologies encounter three significant challenges: 1. Their primary focus on addressing local reasoning, overlooking the intricate underlying cognitive structures of human mind. 2. Numerous prompts are custom-designed for specific structured reasoning processes, neglecting the complexities inherent in real-world natural language tasks. 3. The foundational basis for most prompt designs predominantly relies on intuition, lacking a consolidated epistemological perspective. To address these problems, we turn to epistemology in philosophy.

In the seminal work "Critique of Pure Reason" (Kant et al., 1934), Immanuel Kant delineated an epistemological hierarchy within the human mind: sensibility, understanding, and reason, as depicted in Figure 2. Sensibility constitutes the initial phase, wherein perceptions are amassed as raw sensory impressions, devoid of structured meaning. Subsequently, the stage of understanding interprets these impressions, bestowing structure and significance upon them. Reason stands as the zenith of cognitive capabilities, striving to unearth truths transcending empirical experience. Notably, Kant emphasized that reason must remain rooted in experience to avert "illusions," signifying logical but unrealistic knowledge. Although Kant's philosophy has cast a profound influence across various academic disciplines (Allison, 2004; Bird, 2013; Strawson & Allais, 2018), discussions regarding its application within LLMs are notably scant.

Inspired by Kant's transcendental philosophy (Kant et al., 1934), we propose a prompt framework to augment the performance of LLMs, as illustrated in Figure 1. Our methodology unfolds in four sequential stages: Understand, Plan, Act, and Reflect. Succinctly, the "Understand" stage extracts crucial information from the input text, utilizing a priori human cognitive categories. The "Plan" stage devises an action plan meticulously tailored to the specific task at hand. The "Act" stage embodies the direct problem-solving process of the LLM. Lastly, the "Reflect" phase enables the LLMs to provide feedback and refine its output based on the information above. This framework allows LLMs to generate a multi-level reasoning process and improve its reasoning accuracy by simulating the cognitive ability of the human mind. In addition, such a reasoning process allows humans to correct and provide feedback by checking the output of LLMs at different stages. Finally, the UPAR framework is compatible with various existing prompting techniques, providing possibilities for future systematic prompting engineering.

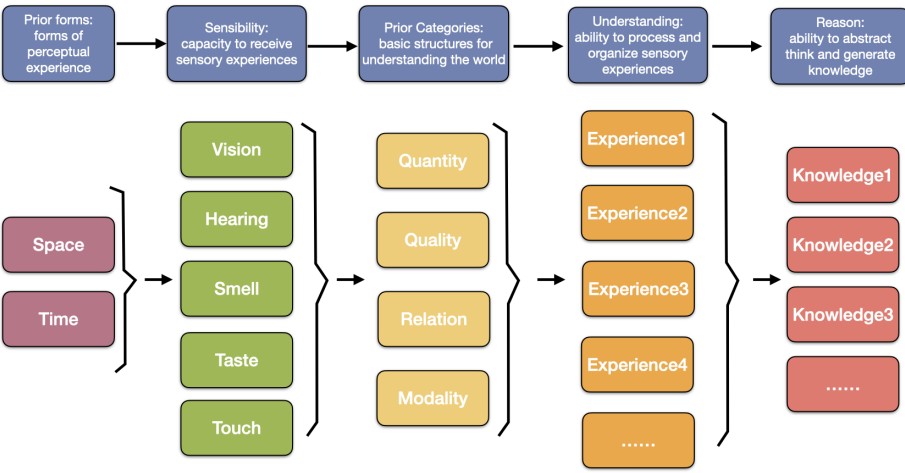

Figure 2: The structure of human mind in Kant's transcendental philosophy

## 2 RELATED WORKS

Our prompting framework draws inspiration from Kant's transcendental philosophy. Therefore in this section, we provide a succinct review of the existing research on prompts and introduce the transcendental principles posited by Kant, elucidating potential linkages between them.

### 2.1 CURRENT PROMPT RESEARCH

#### 2.1.1 INDEPENDENT OF EXTERNAL TOOLS

Numerous studies on prompts have augmented the capabilities of LLMs by delving into single or multi-step thought processes. The Chain-of-Thought (COT) series obliges LLMs to expound the underlying rational processes (Wei et al., 2023; Kojima et al., 2023; Zhang et al., 2022; Fu et al., 2023). Wang et al. (2023b) introduces Self-Consistency, a strategy wherein the LLM iteratively answers the same query, employing majority voting to diminish answer uncertainties. Zhou et al. (2023a) unveils LEAST-TO-MOST, deconstructing intricate problems into sequential sub-tasks. Building upon these foundations, subsequent research has expanded the paradigm to encompass multi-path inferential frameworks (Dohan et al., 2022). Significant advancements include the Tree of Thoughts (Yao et al., 2023a), Graph of Thoughts (Besta et al., 2023), and Algorithm of Thoughts (Sel et al., 2023). These works conceptualize individual LLM computations as fundamental nodes within potential cognitive trajectories, exploring a more extensive spectrum of cognitive pathways.

Another direction involves utilizing the internal knowledge of LLMs to assist decision-making or facilitate automatic optimization. Works by Shwartz et al. (2020); Paranjape et al. (2021); Liu et al. (2022) leverage LLMs to autonomously generate problem-related insights that aid in problem resolution. Jung et al. (2022) proposes the MAIEUTIC, constructing a recursive explanation tree and confining inference within the bounds of a satisfiability problem. Creswell et al. (2022) puts forth the Selection-Inference model, delineating each reasoning step into selection and inference. Madaan et al. (2023) permits LLMs to operate iteratively, alternating between outputting answers and evaluations, and refining. Xie et al. (2023) empowers LLMs to conduct fine-grained self-assessments to enhance multi-step reasoning. Additionally, Yang et al. (2023a); Zhou et al. (2023b) delve into the potential of employing LLMs for the automatic generation and optimization of prompts.

These methods typically yield substantial advancements for LLMs dealing with zero-shot or low-shot prompts. However, several limitations persist. Primarily, these studies often concentrate exclusively on local and specific reasoning processes, neglecting the intrinsic human cognitive structures underpinning language. Concurrently, these manually constructed methods often rely more on intuition than established principles, posing challenges to systematic knowledge in prompt engineering.

#### 2.1.2 USING EXTERNAL TOOLS

Another exploration involves combining LLMs with external tool using. As an intermediary, LLMs have the potential to meld the flexibility of natural language systems with the precision of symbolic language systems, translating natural language problems into the language of external tools. Works in this direction include SEQZERO (Yang et al., 2022), Rational Meaning Construction (Wong et al., 2023), Faithful Chain-of-Thought (Lyu et al., 2023), Program of Thoughts (Chen et al., 2022), Program-aided Language Models (Gao et al., 2023a). The broader concept of tools includes search engines and knowledge databases and even other neural networks, as demonstrated by the tool library (Paranjape et al., 2023), React (Yao et al., 2023b; Yang et al., 2023b), LATM (Cai et al., 2023), Toolformer (Schick et al., 2023), HuggingGPT (Shen et al., 2023), Chameleon (Lu et al., 2023a), FACTOOL (Chern et al., 2023).

These methods integrate LLMs with external tools to augment their capabilities, achieving state-of-the-art results across various tasks. However, they overlook the fact that these tools are products of human thought, not the foundation of thinking. For instance, the human brain, as a large and highly complex network, does not rely on programming languages to solve basic computational problems, nor does it exhibit signs of internal formal reasoning systems. These tools are the creations of human intellect rather than the basis of human reliable thinking. Therefore, in this paper, we exclusively focus on prompting techniques without external tools. As we observe, tool utilization is not a precondition for the proper functioning of the human mind, but merely a supplementary aid.

## 2.2 VIEW FROM PHILOSOPHY

### 2.2.1 RATIONALISM

Rationalism in epistemology posits that knowledge fundamentally emanates from logical reasoning (Markie & Folescu, 2004; Gottlieb & Russell, 2013), frequently neglecting the depth and richness of experiential knowledge. This unwavering perspective subsequently led to the emergence of dogmatism, asserting human rationality as infallible and dismissing the indispensable need for experiential insights. This stance obstructs a more holistic comprehension of the evolution of knowledge. Analogously, early AI models, such as Symbolic AI and Expert Systems (Nilsson, 1982; Luger, 2005), anchored in this rationalistic framework, encountered significant challenges with real-world complexities. Gradually, AI has transitioned from this rigid approach to a more adaptive, data-driven paradigm, exemplified by the rise of deep learning (LeCun et al., 2015).

### 2.2.2 EMPIRICISM

Empiricism posits that all human knowledge originates from sensory experience (Sellars et al., 1956; Markie & Folescu, 2004), asserting that knowledge hinges on external observations rather than intrinsic principles. This perspective, however, can inadvertently steer us towards skepticism, where fundamental notions such as cause and effect might merely represent mental association rather than objective reality, which nullifies the essential possibility of humans discerning the truth and contradicts the scientific system.Empiricism is manifested in machine learning and deep learning, disciplines that prioritize data-driven learning over established rules. Despite the success of these models, their unpredictability and susceptibility to incomprehensible errors, such as adversarial vulnerability (Madry et al., 2017) and LLMs' hallucinations (Rawte et al., 2023; Ji et al., 2023), remain prominent. We have also observed that large language models, rooted in empiricism, suffer from difficulties similar to Hume's problem (Hume, 1875): the lack of genuine understanding regarding the inner laws of the world. LLMs' reasoning rests upon patterns and statistical relationships, devoid of true causality (Hobbhahn et al., 2022; Willig et al., 2022; Zečević et al., 2023; Kıcıman et al., 2023). This observation compels us to explore a more comprehensive epistemological framework to alleviate the prevailing issues faced by current LLMs.

### 2.2.3 KANT'S TRANSCENDENTAL PHILOSOPHY

Kant's transcendental philosophy is dedicated to resolving the contradiction between rationalism and empiricism. In "Critique of Pure Reason" (Kant et al., 1934), Kant constructs an intricate epistemological framework that has fundamentally shaped modern philosophy as we have shown in Fig. 2. Sensibility lies as the first layer of cognitive capacity. Kant advances that sensibility is the initial stage in acquiring perceptions, acting as the fundamental channel between the external world and our cognitive faculties. This perception initially appears as unprocessed sensory data, lacking intrinsic structure or meaning. Kant's delineation of two pure forms of sensibility, space and time, summarizes the transcendental structures we superimpose on experience.

Understanding, as the second layer, epitomizes our capacity to interpret and ascertain meaning from the raw sensory information. It imparts structure and meaning to our experiences. Kant contends that there exist fundamental concepts or "categories" that are intrinsically pure, not emerging from experiential contexts, but rather forming the structural framework we impose upon our experiences. These a priori categories encompass fundamental relationships as is shown in Table 1, forming the bedrock for our interpretation and comprehension of the external realm.

Reason transcends sensibility and understanding, representing the apex of cognitive capabilities. It is devoted to elucidating truths that surpass sensibility and understanding. Kant postulates that reason, grounded on insights furnished by understanding, continuously aspires for systematic, holistic, and ultimate elucidations of existential entities.

Kant's critique also highlights the need for caution and self-reflection in the practice of reason. Despite reason's intrinsic drive to unravel ultimate truths, Kant warns against its venture beyond the boundaries of empirical experience. Kant points out that using reason beyond the confines of experience could produce "illusions", knowledge that seems reasonable but is unrealistic, and advocated rigorous critical examination of our cognitive processes.

Table 1: Transcendental Categories. In Kant's philosophy, the Transcendental Categories are fundamental concepts or pure categories of understanding that structure human cognition and experience. These categories do not derive from acquired experience, but rather serve as an a priori ability to construct possible understanding.

| Category | Subcategories | | |
|---|---|---|---|
| Sensibility | Time | Space | - |
| Quantity | Unity | Plurality | Totality |
| Quality | Reality | Negation | Limitation |
| Relation | Inherence-Subsistence | Causality-Dependence | Community |
| Modality | Possibility | Existence | Necessity |

## 3 METHOD

Based on the analysis above, we generally follow the cognitive structure proposed by Kant's philosophy to build the UPAR (Understand, Plan, Act, Reflect) prompt framework. In the following, we denote a pre-trained LLM with parameters as $p_\theta$ and a sentence as $x$ where $x = (x[1], \ldots, x[n])$, where each $x[i]$ stands for a token. The probabilistic language model can be described as

$$p_\theta(x) = \prod_{i=1}^{n} p_\theta(x[i] \mid x[1, \ldots, i-1]) \tag{1}$$

We further denote the prompting: $p_\theta(\text{output} \mid \text{prompt}(\text{input}))$ as $p_\theta^{\text{prompt}}(\text{output} \mid \text{input})$, where $\text{prompt}(\text{input})$ is the input wrapped with prompt instructions.

### 3.1 UNDERSTAND

In Kant's epistemological framework, the a priori categories of understanding serve the process of human understanding of the world, and are the nodes on the web of knowledge that help us understand natural phenomena (Hegel, 2010). It is these categories that empower us to organize and interpret this raw sensory data. From this perspective, we designate the initial phase of LLM's problem processing as "understanding"—a restoration of a real-world model through the linguistic representation. Specifically, drawing upon Table 1, we prompt LLMs to first address these four questions *within the context of time and space*:

> *Quantity: What entities/events and their quantitative relationships are related to the question?*
>
> *Quality: What intrinsic properties and external constraints of these entities/events are related to the question?*
>
> *Relation: What is the relationship between these entities/events?*
>
> *Modality: Is possibility/impossibility, inevitable/accidental involved in the entities/events related to the question?*

We aim to assist the model better understand entities and their relationships by these questions, especially in the context of time and space. Consider reading a detective fiction: it is crucial to discern characters, objects, events, their attributes, and spatio-temporal coordinates, and comprehend their interactions and relationships. These hidden understandings do not always take the form of language, but allow for more informed conclusions rather than intuitive judgments. Moreover, Shi et al. (2023) affirmed that even slight irrelevant information in queries can notably detract LLMs' performance, highlighting the necessity for information sifting. Hence, we set the first step of the LLM's work as constructing a world model using language. Denoting $u$ as the understanding, this process can be formally written as

$$u \sim p_\theta^{\text{Understand}}(u \mid x) \tag{2}$$

### 3.2 PLAN

In Kant's epistemological framework, reason is situated at the zenith. Reason's capacity lies in processing knowledge that the intellect has already refined, and subsequently integrating this into a systematic comprehension. In this context, we define "reason" as the ability to devise a high-level plan for a target task, grounded on the original context and the structured knowledge extracted by the intellect. We also observe that Bubeck et al. (2023) highlights the limitation imposed by the autoregressive nature of transformer models, as LLMs are constrained in their foresight to "plan ahead" for complex tasks. This notion resonates with the delineation between System 1 and System 2 models in psychology (Kahneman, 2011), where LLMs, functioning as unidirectional word-by-word generative models, inherently lack the "slow thinking" component essential for overseeing the thought process. Consequently, we design the planning stage to compel the model to generate a high-level plan $l$ before embarking on problem-solving, which is prompted and formalized as:

$$l \sim p_\theta^{\text{Plan}}(r \mid x, u), \; \textit{Plan: Let's make a briefly plan to solve this question step by step.} \tag{3}$$

### 3.3 ACT

In the action phase, we prompt LLMs to perform specific calculations or reasoning in accordance with the plan. Ensuring that large language models reason correctly has been a focal point of extensive research in recent years. As previously introduced, numerous works have been dedicated to this area such as Chain-of-Thought (COT) (Wei et al., 2023), Self-Consistency (Wang et al., 2023b), Self-Refine (Madaan et al., 2023). Concurrently, certain studies focus on constructing sufficiently complex LLM thinking paths to fully explore the possible solution space of the problem such as the Tree of Thoughts (Yao et al., 2023a), Graph of Thoughts (Besta et al., 2023), and Algorithm of Thoughts (Sel et al., 2023). We posit that these methodologies can be seamlessly integrated into the action phase of our framework to solve specific problems. Denoting $a$ as the action process, we have

$$a \sim p_\theta^{\text{Act}}(a \mid x, u, l), \; \textit{Act: Now, let's execute the plan step by step.} \tag{4}$$

### 3.4 REFLECT

In Kant's critical philosophy, an important aspect is the examination of reason within its own confines: the application of reason must not deviate from experience. Exceeding these bounds may lead reason astray, giving rise to erroneous "illusions". This notion resonates with observations made in LLMs' hallucinations (Kadavath et al., 2022; Liang et al., 2022; McKenna et al., 2023; Li et al., 2023; Ji et al., 2023; Rawte et al., 2023). Therefore, we propose the final step as reflection, requiring LLMs to comprehensively review the content they generate, point out possible errors, and correct them. Many recent methods employ similar self-checking concepts to mitigate the LLMs' hallucinations, such as Self-Consistency (Wang et al., 2023b), Self-Refine (Madaan et al., 2023), DOLA (Chuang et al., 2023), citation generation (Gao et al., 2023b), Retrieval Augmentation (Ren et al., 2023). Essentially, these techniques can be used to enhance the "reflect" stage in our UPAR framework. Denoting $r$ as the reflection, "reflect" is prompted and formalized as:

$$r \sim p_\theta^{\text{Reflect}}(r \mid x, u, l, a), \; \textit{Reflect: Check your answers and correct possible errors.} \tag{5}$$

### 3.5 UPAR FRAMEWORK

$$(u, l, a, r) \sim \underbrace{p_\theta^{\text{Understand}}(u \mid x)}_{\text{Understand}} \cdot \underbrace{p_\theta^{\text{Plan}}(l \mid x, u)}_{\text{Plan}} \cdot \underbrace{p_\theta^{\text{Act}}(a \mid x, u, l)}_{\text{Act}} \cdot \underbrace{p_\theta^{\text{Reflect}}(r \mid x, u, l, a)}_{\text{Reflect}} \tag{6}$$

Based on the aforementioned discussion, we hereby formalize the entire UPAR framework as outlined in Eq 6. For the scalability of our approach, we provide all these prompts as system-level inputs to the model in a single instance, which means we do not provide any few-shot samples, do not manually split any sub-problems, and do not connect or modify the model output. Specifically, we force the model to do all understanding, planning, execution, and reflection in a single output to verify to a certain extent whether the large language model has human-like system thinking capabilities. An example from GSM8K is shown in Figure 3.

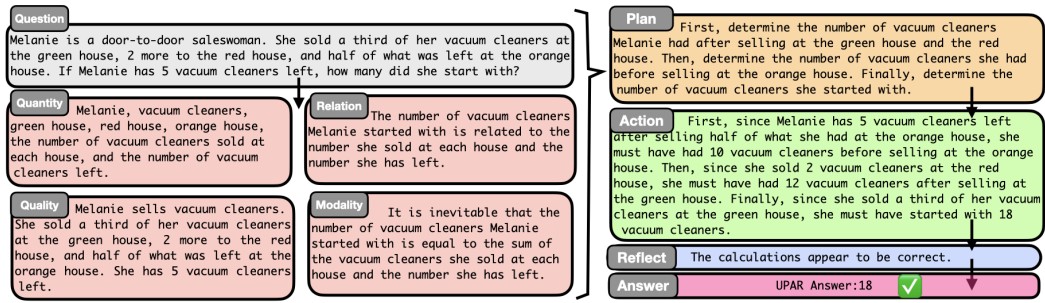

Figure 3: An UPAR framework example from GSM8K, created by GPT4.

# 4 RESULTS

## 4.1 EXPERIMENT SETTING

We choose SCIBENCH (Wang et al., 2023a), GSM8K (Cobbe et al., 2021), AQUA-RAT (Ling et al., 2017), CommonsenseQA (Talmor et al., 2019), StrategyQA (Geva et al., 2021), and the Causal-Judgement task in BIG-Bench Hard (BBH) (Suzgun et al., 2022) as the test datasets. SCIBENCH is a recent challenging dataset contains collegiate-level scientific mathematics, chemistry, and physics problems. Further, we manually selected questions from GSM8K that could not be answered correctly by zero-shot Chain-of-Thought (COT) method (Kojima et al., 2022) on GPT4. After manual inspection and removal of incorrectly labeled data, 48 items were retained to form the GSM8K-Hard (GSM8K-H) dataset. Both SCIBENCH and GSM8K-H require significantly more complex reasoning and understanding processes. We evaluated our method using the GPT4 (OpenAI, 2023) models; all temperatures were set to 0 and top_p to 1. For UPAR, we test without any external tools, using it as only the one-time zero-shot system-level prompt, which means we do not provide manually constructed few-shot samples; do not perform multiple Q&A iterations; do not manually divide sub-questions, and only use the entire contents of model's single output as the final answer.

## 4.2 MAIN RESULTS

Table 2: GPT4 model accuracy (%) on SCIBENCH textbook dataset. We take baseline methods and data from (Wang et al., 2023a): zero-shot learning without the system prompt (*Zero−S*), zero-shot learning with the system prompt (*Zero*), few-shot learning (*Few*), CoT prompting under zero-shot (*Zero+CoT*) and few-shot learning (*Few+CoT*) scenarios, few-shot learning that prompts to use Python (*Few+Py*), and Wolfram Language (*Few+Wol*) as external tools. The best performing score is highlighted in **bold** and second-best is underlined.

| Setting | Chemistry | | | | Physics | | | Math | | | Avg. |
|---|---|---|---|---|---|---|---|---|---|---|---|
| | atkins | chemmc | quan | matter | fund | class | thermo | diff | stat | calc | |
| Zero−S | 14.95 | 25.64 | 8.82 | 18.37 | 21.92 | 12.77 | 7.46 | 8.00 | 28.00 | 19.05 | 16.81 |
| Zero | 27.10 | 23.08 | 14.71 | 22.45 | 15.07 | 8.51 | 11.94 | 18.00 | 56.00 | 42.86 | 25.09 |
| Zero+CoT | 28.04 | 43.59 | 14.71 | 20.41 | 21.92 | **19.15** | 17.91 | 22.00 | 50.67 | 42.86 | 28.52 |
| Few | 15.87 | 30.77 | 17.65 | 12.24 | 26.03 | 12.77 | 5.97 | 8.00 | 49.33 | 33.33 | 21.46 |
| Few+CoT | 21.05 | 46.15 | 17.65 | 26.53 | 27.40 | 14.00 | 13.43 | 18.00 | 61.33 | 35.71 | 28.35 |
| Few+Py | 21.05 | 41.03 | **38.24** | 28.57 | 38.36 | 17.02 | 29.85 | 34.00 | 69.33 | 42.86 | 35.80 |
| Few+Wol | 3.74 | 0.00 | 17.65 | 26.53 | 27.30 | 17.02 | 17.91 | 32.00 | 7.69 | 14.29 | 15.56 |
| UPAR | **55.28** | **62.5** | 35.71 | **42.37** | **66.26** | 18.52 | **34.52** | **38.18** | **70.53** | **61.54** | **50.65** |

Results of SCIBENCH in Table2 demonstrate that UPAR shows consistent superiority and stability over existing prompting methods on complex reasoning task. In particular, it can be noted that on this dataset: 1. Few-shot learning has only a weak performance improvement compared to zero; 2.

The use of Python and Wolfram ensures that LLM has reliable numerical and symbolic calculation tools, but it also does not significantly improve the reasoning ability. In contrast, the UPAR method consistently achieved the best results without relying on any few-shot examples or using external tools. For such scientific problems that require complex reasoning processes, we argue that the a priori category context and multi-level cognitive framework in UPAR, instead of linear thinking process, can significantly increase the reasoning capabilities of LLMs.

| Method | GSM8K | AQuA | CSQA | Strategy | GSM8K-H | Causal-Judgement |
|--------|-------|------|------|----------|---------|------------------|
| Zero | 39.12 | 43.70 | **84.6** | 67.78 | 16.67 | 70.05 |
| Zero+CoT | 93.10 | 74.41 | 81.16 | 72.76 | 22.92 | 67.91 |
| UPAR-S | **94.26** | **75.98** | 84.03 | **73.63** | **58.33** | 72.73 |
| UPAR | 93.10 | **75.98** | 83.13 | 68.90 | 47.92 | **75.40** |

Table 3: Performance of models on arithmetic and commonsense data with GPT4

We also show experiment results on GSM8K, AQUA, CommonsenseQA, and StrategyQA and Causal-Judgement task in Table 3. Since most of the questions in these dataset are relatively simple and do not involve complex cognitive processes, we further designed UPAR-Simple, a simplified version of the UPAR. The only difference is the level of detail asked in the understanding step: in UPAR-S , we only asked "the entities and relationships" to avoid introducing irrelevant content.

As anticipated, the UPAR-Simple (UPAR-S) model consistently outperforms other models in the majority of the tasks. It notably achieves a score of 58.33% on the GSM8K-H task, a significant margin higher compared to the COT baseline of 22.92%. This evidence underscores the robustness and superior efficacy of UPAR-S in tackling various tasks. In the Causal-Judgment task, problems typically have complex backgrounds involving multiple tasks and the judgment of causal relationships across various times. As expected, UPAR showcases exceptional performance in the Causal-Judgment task, attaining the highest score of 75.40%, surpassing the COT baseline of 67.91%. This result highlights the significant capability improvement of UPAR in complex reasoning tasks, demonstrating the rationality of our proposed multi-level cognitive architecture.

### 4.3 ABLATION STUDY ON UPAR

In the ablation study presented in Table 4, we remove different parts of UPAR-S and test their performance differences using GPT-4 on the GSM8K-Hard dataset. The complete UPAR-S model achieves an accuracy of 58.33%. When the understanding component is removed, the accuracy slightly decreases to 54.17%. Furthermore, the model without the planning component has an accuracy of 56.25%, lower than the full model but still showing some competence. After removing the action component, the model's performance drastically drops to an accuracy of only 16.67%. Lastly, without the reflection component, the accuracy is 43.75%, representing a substantial decline compared to that of the complete model. These data underscore the critical role of each component in the UPAR model. We further tested the stability of the UPAR method at different temperatures, as demonstrated in Figure 4. Experimental results show that UPAR stably outperforms COT at various temperatures in terms of inference performance.

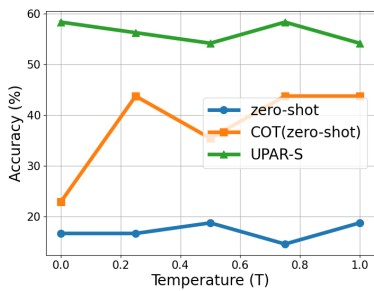

Figure 4: GPT4 performance of zero-shot, COT and UPAR-S on GSM8K-Hard versus temperature.

Table 4: Ablation study on GSM8K-Hard, with GPT4, UPAR-S method.

| Method | Accuracy | Method | Accuracy | Method | Accuracy |
|--------|----------|--------|----------|--------|----------|
| Zero-shot | 16.67% | w/o understand | 54.17% | w/o action | 16.67% |
| UPAR-S | **58.33**% | w/o plan | 56.25% | w/o reflect | 43.75% |

## 4.4 ERROR ANALYSIS

According to our experiments, LLMs under UPAR may make errors in the following situations: 1.Understand: LLMs may misunderstand the ambiguous semantic content and generate a deterministic attitude answer rather than seek clarification. 2.Plan: LLMs may produce illogical plans for solving problems, even under the premise of correctly understanding. 3.Execution: LLMs cannot accurately perform computational problems such as floating-point multiplication, matrix operations, calculus, complex algebraic equations, etc. Without relying on external tools, LLMs often output unreliable results or refuse to calculate. 4.Reflection: LLMs may not be able to effectively correct their own errors without external suggestion, such as modifying the original correct answer into an incorrect one, or being unable to effectively identify it's own erroneous calculations and reasoning processes. We manually analyzed all incorrect answers on the GSM8K-Hard dataset and classified the reasons, as shown in Figure 5. we anticipate the incorporation of other methods into UPAR can further minimize these errors.

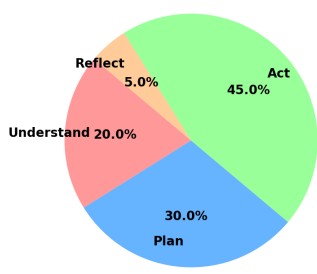

Figure 5: GPT4 UPAR-S method's error source on GSM8K-Hard.

Such methods may include using external knowledge to ensure correct understanding (Lyu et al., 2023), utilizing mathematical tools (Wong et al., 2023) to ensure accurate calculations, introducing multiple rounds of self-refinement (Madaan et al., 2023) to enhance reflection effectiveness.

## 4.5 COMPATIBILITY WITH SELF-REFINE METHOD

In order to assess UPAR's compatibility with existing methods, we further attempted to integrate UPAR with the self-consistency (SC) method (Wang et al., 2023b). The experimental results are shown in Table 5. The self-consistency method improves the responses accuracy by letting LLMs answer the same question multiple times and using a majority vote to determine the final result. Each experiment was conducted five times to conduct SC. It can be observed that the UPAR method is highly compatible with the SC method, achieving significant performance improvement. For instance, on GSM8K-Hard, the combination of UPAR and SC improves the accuracy from 47.92% to 60.42%, while the combination of COT and SC does not significantly improve the performance. This is usually because the question involves some non-linear reasoning process that cannot be effectively captured by linear thinking processes, while UPAR alleviates this significantly.

Table 5: Combinations of different methods with Self-Consistency.

| Method | GSM8K-H | GSM8K-H (SC) | C-J | C-J (SC) |
|---|---|---|---|---|
| Zero-Shot | 16.67% | 18.75% | 70.05% | 72.19% |
| CoT(Zero-Shot) | 22.92% | 18.75% | 67.91% | 72.72% |
| UPAR-S | **58.33**% | **64.58**% | **72.73**% | **73.26**% |
| UPAR | **47.92**% | **60.42**% | **75.40**% | **76.47**% |

# 5 CONCLUSION AND DISCUSSION

This paper aims to fill the existing gap in the epistemological foundation in the field of prompting techniques for Large Language Models (LLMs). Inspired by Kant's philosophy of a priori knowledge, we introduce the UPAR prompting framework, emulating human cognition structures within LLMs. The UPAR framework is divided into four phases: "Understand", "Plan", "Act" and "Reflect", enabling the extraction of structured information from complex contexts, plan solutions in advance, execute according to the plan, and self-reflect. This structure significantly enhances the explainability and accuracy of LLM inference, producing a human-understandable and inspectable inferential trajectory. Furthermore, Kant's a priori philosophical epistemological framework brings a new perspective to the LLMs community, which is currently dominated by empiricism. We believe that integrating these a priori epistemic structures as additional constraints into LLMs training is a valuable topic.

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

APPENDIX

In the appendix we show the complete prompt words and some examples to help readers understand our UPAR method.

# A    FULL SETS OF PROMPTS

## A.1    UPAR-S

You are an assistant with a multi-level thinking structure. When you receive the problem, don't solve it immediately. Follow the structured instructions:

First, let's briefly understand this question step by step.: [Your understanding here.]

Let's make a briefly plan to solve this question step by step: [Your plan here.]

Now, let's execute the plan step by step: [Your solution here.]

Check your answers and correct possible errors.: [Your reflect here.]

## A.2    UPAR

You are an assistant operating under a Kantian-inspired multilevel thinking structure. When receiving a problem, do not solve it immediately. Instead, follow these structured instructions:

First, briefly understand this question in the context of time and space step by step.

Quantity: What entitie/events and their quantitative relationships are related to the question? [Your answer here]

Quality: What intrinsic properties and external constraints of these entities/events are related to the question? [Your answer here]

Relation: What is the relationship between these entities/events? [Your answer here]

Modality: Is possibility/impossibility, inevitable/accidental involved in the entities/events related to the question? [Your answer here]

Let's make a briefly plan to solve this question step by step: [Your plan here.]

Now, let's execute the plan step by step: [Your solution here.]

Check your answers and correct possible errors.: [Your reflect here.]

These multi-step thinking processes can be provided to LLMs at once using the ["role":"system"] part of OpenAI's API reference here, or can be asked step by step in multiple rounds of dialogue. We expect the latter to have better results, but for efficiency reasons, we use the former in our experiments. In all experiments, we use the zero-shot approach, i.e., no samples are manually constructed.

# B EXAMPLES

## B.1 GSM8K

---

**Correct example 1**

**Question:** Bill is ordering a new truck. He has decided to purchase a two-ton truck with several added features: a king cab upgrade, a towing package, leather seats, running boards, and the upgraded exterior light package. The base price of the truck is $30,000, and the other features are at extra cost. The king cab is an extra $7,500, leather seats are one-third the cost of the king cab upgrade, running boards are $500 less than the leather seats, and the upgraded exterior light package is $1500. What is the total cost of Bill's new truck, in dollars?

**Answer:**

**Understand**

*Quantity*: The entities related to the question are the base price of the truck ($30,000), the cost of the king cab upgrade ($7,500), the cost of the leather seats (one-third the cost of the king cab upgrade), the cost of the running boards ($500 less than the leather seats), and the cost of the upgraded exterior light package ($1500).

*Quality*: The intrinsic properties related to the question are the costs of the truck and its features. The external constraint is that these costs need to be added together to find the total cost of the truck.

*Relation*: The relationship between these entities is that they are all costs that need to be added together to find the total cost of the truck.

*Modality*: The total cost of the truck is inevitably the sum of the costs of the truck and its features.

**Plan**

First, calculate the cost of the leather seats by dividing the cost of the king cab upgrade by three.
Then, calculate the cost of the running boards by subtracting $500 from the cost of the leather seats.
Finally, add the costs of the truck, the king cab upgrade, the leather seats, the running boards, and the upgraded exterior light package to find the total cost of the truck.

**Act**

The cost of the leather seats is $7,500 / 3 = $2,500.
The cost of the running boards is $2,500 - $500 = $2,000.
The total cost of the truck is $30,000 + $7,500 + $2,500 + $2,000 + $1,500 = $43,500.

**Reflect**

The calculations appear to be correct, and the total cost of the truck is reasonable given the costs of the truck and its features. Therefore, I believe the answer is correct.

**Answer: 43500, GroundTruth: 43500**

---

**Correct example 2**

**Question:** At the trip to the county-level scavenger hunt competition, 90 people were required to split into groups for the competition to begin. To break people up into smaller groups with different leaders, 9-person groups were formed. If 3/5 of the number of groups each had members bring back 2 seashells each, how many seashells did they bring?

**Answer:**

**Understand**

*Quantity*: The entities related to the question are the 90 people, the 9-person groups, the 3/5 of the groups, and the 2 seashells each person brought back.

*Quality*: The intrinsic properties related to the question are the number of people in each group, the fraction of groups that brought back seashells, and the number of seashells each person brought back.

*Relation*: The relationship between these entities is that the 90 people are divided into 9-person groups, and 3/5 of these groups each brought back 2 seashells.

*Modality*: The events described in the question are inevitable as they are stated as facts.

**Plan**

First, calculate the total number of groups by dividing the total number of people by the number of people in each group.

Then, calculate the number of groups that brought back seashells by multiplying the total number of groups by 3/5.

Finally, calculate the total number of seashells brought back by multiplying the number of groups that brought back seashells by the number of seashells each person brought back and the number of people in each group.

**Act**

First, calculate the total number of groups: 90 people / 9 people/group = 10 groups.

Then, calculate the number of groups that brought back seashells: 10 groups * 3/5 = 6 groups.

Finally, calculate the total number of seashells brought back: 6 groups * 9 people/group * 2 seashells/person = 108 seashells.

**Reflect**

The calculations appear to be correct, and the total cost of the truck is reasonable given the costs of the truck and its features. Therefore, I believe the answer is correct.

**Answer: 108, GroundTruth: 108**

---

**Wrong example 1: wrong reflection**

**Question:** A company pays each of its employees $600 in a month. The company has a policy of increasing the salaries of each of its employees by 10% of the initial salary every year for those who've stayed in the company for five years. If Sylvie just clocked 5 years in the company last December, what's her annual salary after three more years of service?

**Answer:**

**Understand**

The question is asking for the annual salary of an employee named Sylvie after three more years of service, given that she has already worked for the company for five years. The company pays each of its employees $600 per month and increases the salaries by 10% every year for those who've stayed in the company for five years.

**Plan**

1. Calculate Sylvie's current annual salary.

2. Calculate the annual salary increase.

3. Apply the increase for the next three years.

**Act**

1. Sylvie's current annual salary is $600 * 12 = $7200

2. The annual salary increase is 10% of $7200 = $720.

3. After three more years, her salary will be increased by $720 * 3 = $2160.

4. So, Sylvie's annual salary after three more years of service will be $7200 + $2160 = $9360.

**Reflect**

The calculation seems correct, but upon reflection, I realized that the 10% increase is compounded annually, not added up linearly. This means that each year, the increase is calculated based on the new salary, not the original one. So, the correct calculation should be:

1. After the first year, Sylvie's salary will be $7200 + 10% of $7200 = $7920.

2. After the second year, her salary will be $7920 + 10% of $7920 = $8712.

3. After the third year, her salary will be $8712 + 10% of $8712 = $9583.2.

So, Sylvie's annual salary after three more years of service will be $9583.2.

**Answer: 9583.2, GroundTruth: 9350**

**Wrong example 2: wrong action**

**Question:** Twenty dozen cups cost $1200 less than the total cost of half a dozen plates sold at $6000 each. Calculate the total cost of buying each cup.

**Answer:**

**Understand**

The problem is asking us to find the cost of each cup. We know that twenty dozen cups cost $1200 less than the total cost of half a dozen plates sold at $6000 each.

**Plan**

1. Calculate the total cost of half a dozen plates.

2. Subtract $1200 from the total cost of the plates to find the total cost of the cups.

3. Since we know that twenty dozen cups were bought, we can find the cost of each cup by dividing the total cost of the cups by the total number of cups.

**Act**

1. The total cost of half a dozen plates is 6/2 * $6000 = $18000.

2. The total cost of the cups is $18000 - $1200 = $16800.

3. Twenty dozen cups is 20 * 12 = 240 cups.

4. Therefore, the cost of each cup is $16800 / 240 = $70.

**Reflect**

The cost of each cup is $70. This answer makes sense given the information provided in the problem.

1. After the first year, Sylvie's salary will be $7200 + 10% of $7200 = $7920.

2. After the second year, her salary will be $7920 + 10% of $7920 = $8712.

3. After the third year, her salary will be $8712 + 10% of $8712 = $9583.2.

So, Sylvie's annual salary after three more years of service will be $9583.2.

**Answer: 9583.2, GroundTruth: 9350**

## B.2 CAUSAL-JUDGMENT

---

**Correct example 1**

**Question:** How would a typical person answer each of the following questions about causation? A machine is set up in such a way that it will short circuit if both the black wire and the red wire touch the battery at the same time. The machine will not short circuit if just one of these wires touches the battery. The black wire is designated as the one that is supposed to touch the battery, while the red wire is supposed to remain in some other part of the machine. One day, the black wire and the red wire both end up touching the battery at the same time. There is a short circuit. Did the black wire cause the short circuit?

**Answer:**

**Understand**

*Quantity*: The entities involved in this question are a machine, a black wire, a red wire, a battery, and a short circuit. The event is the simultaneous touching of the battery by both wires, leading to a short circuit.

*Quality*: The machine is designed in such a way that it will short circuit if both wires touch the battery simultaneously. The black wire is supposed to touch the battery, while the red wire is not. The short circuit is a result of both wires touching the battery at the same time.

*Relation*: The relationship between these entities is that the black wire and the red wire are part of the machine and their interaction with the battery can cause a short circuit.

*Modality*: The short circuit is inevitable if both wires touch the battery at the same time. It is accidental that the red wire also touched the battery, leading to the short circuit.

**Plan**

The plan is to analyze the causation of the short circuit based on the information provided.

**Act**

A typical person might answer that the black wire did not cause the short circuit. The short circuit was caused by both the black and red wires touching the battery at the same time. The black wire was supposed to touch the battery as part of its normal operation, so it did not cause the short circuit. The red wire, which was not supposed to touch the battery, did so, and this unexpected event, in combination with the black wire touching the battery, caused the short circuit.

**Reflect**

The analysis seems correct based on the information provided. The short circuit was caused by both wires touching the battery simultaneously, not just the black wire.

**Answer: No, GroundTruth: No**

---

---

**Correct example 2**

**Question:**How would a typical person answer each of the following questions about causation? Claire's parents bought her an old computer. Claire uses it for schoolwork, but her brother Daniel sometimes logs on to play games. Claire has told Daniel, "Please don't log on to my computer. If we are both logged on at the same time, it will crash". One day, Claire and Daniel logged on to the computer at the same time. The computer crashed. Later that day, Claire's mother is talking with the computer repairman. The repairman says, "I see that Daniel was logged on, but this computer will only crash if two people are logged on at the same time. So, I still don't see quite why the computer crashed." Did Daniel cause the computer crash?

**Answer:**

**Understand**

*Quantity*: The entities involved in this question are Claire, Daniel, the old computer, and the computer repairman. The event is the computer crashing. The quantitative relationship is the number of people logged on to the computer at the same time.

*Quality*: The intrinsic properties related to the question include the old computer's limited capacity to handle multiple users at the same time, and Daniel's disregard for Claire's warning. The external constraints include the computer repairman's understanding of the situation.

*Relation*: The relationship between these entities is that Claire and Daniel are siblings who use the same computer, and the computer repairman is trying to understand why the computer crashed.

*Modality*: The possibility of the computer crashing is related to the number of people logged on at the same time. The crash is inevitable if two people are logged on at the same time, and accidental if only one person is logged on.

**Plan**

To answer the question, we need to consider the information provided and apply a logical analysis. We know that the computer crashes when two people are logged on at the same time, and that both Claire and Daniel were logged on when the computer crashed. Therefore, we can infer that Daniel's action of logging on to the computer contributed to the crash.

**Act**

Yes, Daniel did cause the computer to crash. Although he was not the only one logged on at the time, his action of logging on despite Claire's warning contributed to the situation that caused the computer to crash.

**Reflect**

The answer seems logical and consistent with the information provided in the question. However, it's important to note that causation is often complex and can involve multiple factors. In this case, both Claire and Daniel's actions contributed to the computer crash.

**Answer: Yes, GroundTruth: Yes**

---

### B.3 UPAR AS AN EMERGENT ABILITY

We further tested UPAR on GPT-3.5, as shown in Table 6, and plotted the difference in accuracy between UPAR and COT in Figure 6(a)(b). Different from GPT-4, experimental data on GPT-3.5 show that UPAR obtains slightly weaker results than COT in most cases. Consistent with our intuition, this implies that UPAR, as a more complex higher-level thinking framework, is an emergent ability (Wei et al., 2022). Essentially, autoregressive transformer-based LLMs are suitable for generating continuous sequences of text, especially when such linguistic associations occur repeatedly in the training data. Meanwhile, UPAR, aiming to mimic the structure of human thinking, compels Large Language Models (LLMs) to undergo multi-level, non-linear thinking processes. These potential cognitive processes, typically unrepresented in natural language, are speculated to be burdensome

for weaker models. However, for GPT-4, experiments demonstrate its effective utilization of intermediate results generated at each level of thinking, leading to more accurate answers.

| Method | GSM8K | AQuA | CSQA | Strategy | GSM8K-H | Causal-Judgement |
|--------|-------|------|------|----------|---------|------------------|
| Zero-Shot | 15.85% | 29.53% | **74.12%** | **65.37%** | 6.25% | 57.22% |
| CoT(Zero-Shot) | 75.74% | **55.51%** | 70.60% | 63.19% | 37.5% | **61.50%** |
| UPAR-S | **77.18%** | 50.79% | 68.63% | 62.23% | **39.58%** | 60.96% |
| UPAR | 71.04% | 51.18% | 69.04% | 49.23% | 20.83% | 59.36% |

Table 6: Performance of models on arithmetic and commonsense data with GPT3.5

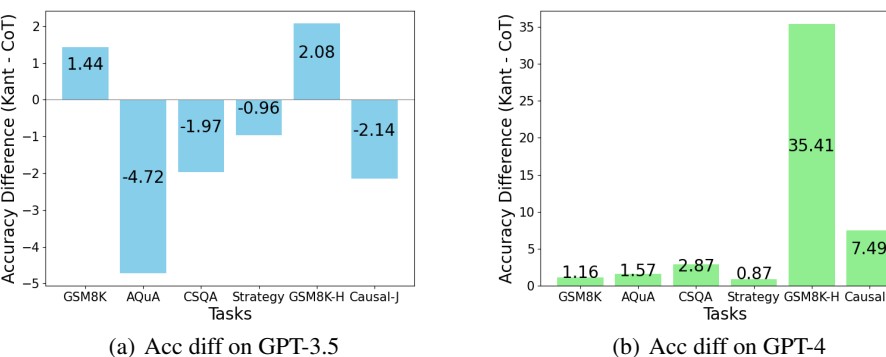

(a) Acc diff on GPT-3.5         (b) Acc diff on GPT-4

Figure 6: (a) GPT3.5 performance on the GSM8K-Hard, UPAR accuracy minus COT accuracy. (b) GPT4 performance on the GSM8K-Hard, UPAR accuracy minus COT accuracy.

