# OpenReview forum: "UPAR: A Kantian-Inspired Prompting Framework for Enhancing Large Language Model Capabilities"
_ICLR.cc/2024/Conference — Submitted to ICLR 2024_

### Official Review · Reviewer_Eftc · 2023-10-31

**Soundness:** 2 fair
**Presentation:** 2 fair
**Contribution:** 2 fair
**Rating:** 3
**Confidence:** 3

**Summary:**

This paper proposes a Kant-inspired framework for the sequential prompting of LLMs. After revisiting Kant's transcendental philosophy structure, the paper proposes a four-step framework that is based on these ideas, consisting of understanding, planning, acting, and reflecting. The newly proposed framework called UPAR is compared to existing methods that roughly cover some but not all aspects of UPAR, and shows to be superior when using GPT-4.

**Strengths:**

S1) The argument that the ongoing paradigm shift brought by LLMs can use some formal framework from cognitive science is compelling and timely.

S2) The idea to synthesize the thinking steps along the four subsequent components is interesting and aligns well with prior work on LLMs.

S3) The results, especially the ablations, are informative and improve the understanding of the contribution of each component.

S4) The formalizations of the four steps are useful and refreshing.

**Weaknesses:**

W1) The chief weakness of this paper is that the paper seems to exaggerate its contribution. The promise of the paper is that it will ground the thinking steps with LLMs in some objective framework that has been well-accepted in psychology/philosophy. What happens in the paper is that the coupling between Kant's theory and the UPAR framework is loose at best (compare figures 2 and 3). Now, the paper is unclear whether UPAR is: A) only inspired by this theory or B) claims to be supported by this theory (which is a much stronger claim).

If A) is the claim, then a loose coupling would be fine, but this would undermine many of the novelty claims (which are anyway difficult to follow), like "often concentrate exclusively on local and specific reasoning processes, neglecting the intrinsic human cognitive structures underpinning language", "these tools are products of human thought, not the foundation of thinking", and "these tools are the creations of human intellect rather than the basis of human reliable thinking".

If B) is what the paper claims, then the authors really need to justify why Kant's theory is taken as the golden standard of "human reliable thinking" and how the UPAR framework aligns seriously with Kant's framework.

W2) While the paper emphasizes the need for a model to receive the full complexity of UPAR thinking, in fact, the main UPAR variant being emphasized is its "simple" variant, which replaces the understanding aspects of Kant's framework with other ones (entities and relations) that are indeed intuitively more useful for the tasks at hand. Surprisingly, the authors do not comment on this finding and what this means for the overall premise of the work.

W3) The paper makes claims that using reasoning would reduce "illusions", which is a nice and compelling statement. However, it is unclear whether UPAR indeed results in less illusions. In general, it is unclear what the qualitative improvement brought by UPAR is; but the improvement does not seem to be some emergent/qualitative jump, but rather a little better overall score while still producing judgments that are as unreliable as the baseline model (as far as I can see, there is no way to guarantee reliability of the reasoning in UPAR).

**Questions:**

Q1) Can you please clarify the relation between Kant's theoretical work and UPAR?

Q2) How is Kant's theory (or UPAR's framework) guaranteeing higher reliability or less illusions?

Q3) How do you interpret the fact that the UPAR-S method is generally much more useful than the UPAR framework? What does this mean for the general premise of the paper?

---

> ### Author Response · Authors · 2023-11-15
> **Thank you for your insightful review!**
>
> Q1: Can you please clarify the relation between Kant's theoretical work and UPAR?
>
> A1: Regarding your question, our paper belongs to A. The UPAR method is inspired by Kant's transcendental philosophy, rather than strictly following it. This means that there is indeed a loose coupling between the two. The biggest difference is that in Kant's philosophy, cognitive categories are inherent cognitive structures of humans, even without exposure to real-world data. However, Transformer, the infrastructure of large language models, obviously does not naturally have these capabilities. To alleviate this contradiction, we propose to explicitly extract these cognitive categories from the trained large-scale language model itself and use them as problem-solving context to enhance the reliability of the inference process.
>
> More importantly, we wish to point out that we are trying to bring Kant's transcendental philosophy into current discussions in the field of large language models, rather than claiming that we provide the only way to achieve it. The core question we hope to answer is whether such cognitive structures can provide performance beyond simple prompting techniques. We particularly look forward to follow-up work exploring this direction, particularly discussing the closer integration of these cognitive categories as a ``pre"-training/inherent ability with neural networks.
>
> Q2: How is Kant's theory (or UPAR's framework) guaranteeing higher reliability or less illusions?
>
> A2: As we admitted for Q1, our approach does not strictly follow Kantian philosophy. The answer proposed by Kant's philosophy to the question of how humans can reliably understand the world is: 1. Human beings have a priori categories that do not rely on empirical data as the basic cognitive structure for constructing empirical data. 2. Human reason can carry out higher-level summarization, reasoning and self-reflection processes based on perceptual experience organized by these categories. And we try to let the LLM simulate a similar process in UPAR.
>
> The key to its ability to reduce LLM illusions is that in traditional prompting techniques, we are not sure whether the LLM can truly "understand" these basic category relationships in complex contexts, and the reasoning process based on unreliable understanding is more prone to errors. In the UPAR framework, we explicitly guide the LLM to generate these basic understandings about the problem, and simulate the planning, execution, and reflection processes of human thinking based on the context. On one hand, this process can force the LLM to recall more structured "world knowledge" to generate a deeper understanding of the problem. On the other hand, it can produce multi-level, understandable, verifiable, and modifiable reasoning trajectories for humans. More precisely, when we say "reduce illusions" we mean "reduce errors for challenging questions that require a more structured human-like reasoning."
>
> Q3: How do you interpret the fact that the UPAR-S method is generally much more useful than the UPAR framework? What does this mean for the general premise of the paper?
>
> A3: We propose the UPAR-S method as a compromise for dealing with simple questions. The reason why we did this is because most of the problems in public datasets are pretty simple  and do not involve complex category relationships.  For example, GSM8K contains many simple questions like:\textbf{``A robe takes 2 bolts of blue fiber and half that much white fiber.  How many bolts in total does it take?"}
>
> In a pragmatic standpoint of view, introducing complex category concepts in answering these simple questions may not only be unnecessary, but could even confuse the LLM. Based on this, we proposed UPAR-S, which follows the main steps of UPAR  but does not prompt the model to understand the question based on Kant's categories.  This flexibility tends to yields better results for simple questions.
>
> On the more complex SCIBENCH, GSM8K-Hard and Causal-Judgement dataset (usually a paragraph of about 100 words, including stories of multiple times, spaces, characters, objects, and causal relationships), our complete UPAR method shows a significant accuracy improvement.  Indeed, these tasks involve more sophisticated  reasoning in a more complex context, for which the full version of UPAR could guide the LLM gain a deeper understanding of the task based on Kant's categories, and hence helps the model better reason about the causal relationships.
>
> We recognize the potential for future enhancement through the creation of a mechanism that strategically employs either UPAR or UPAR-s, guided by an assessment of the question's difficulty level. Implementing such a mechanism would offer a more adaptable and practical solution for diverse problems.
>
> We'd be glad to discuss any unclear details further.

---

> ### Author Response · Authors · 2023-11-19
> **Additional Data and details**
>
> Dear reviewers, please review our general response for additional strong experimental data and details.

---

### Official Review · Reviewer_RawG · 2023-11-02

**Soundness:** 3 good
**Presentation:** 4 excellent
**Contribution:** 2 fair
**Rating:** 6
**Confidence:** 2

**Summary:**

This paper represents UPAR, a prompting framework inspired by Immanuel Kant's arguments about the structure of the human mind. It consists of "understand, plan, act, reflect" steps which ask the model to break down its given problem in a pre-specified way.

**Strengths:**

* The paper was fun to read and draws on interesting ideas.
* The prompting approach is simple and seems like it'd be easy to understand if it were fully described in the paper.

**Weaknesses:**

1. The paper lacks a lot of important details.
  a. I'm confused by the descriptions of the P, A, and R steps. Sections 3.2–3.4 just discuss the philosophical side and motivation without saying how the model is actually prompted. That seems to me like the most important thing to communicate in the paper. Please include it, like you did with "Understand". Also, please be more specific even in the "Understand" section about how your prompting approach works. Do you prompt it four separate times, once for each question? How do you instruct the model outside of just asking it the question?
  b. Does it work only on instruction-tuned or RLHF models, or is it designed to work with pretrained LMs as well? Can you use few-shot examples? Where would they come from? Is there some set of tasks on which it doesn't work? What changes besides the simplification in UPAR-S might be necessary to make it work in other cases?
2. The results are not very promising. It does yield improvements over zero-shot CoT with GPT-4, but only very small ones, and it's unclear whether they are statistically significant (how big are the test sets? What's the total n being tested on?). The only case of a large gap with GPT-4 zero-shot was on a subset of GSM8k _filtered to examples which zero-shot CoT with GPT-4 got wrong_ — not a fair comparison.
3. It's not totally clear to me how deep the relationship goes between the prompting approach and the philosophical backdrop. Especially given the lack of detail in the paper, one can imagine many possible ways of implementing the same idea. Why this way in particular? For example, the assignment of questions to the pure categories of understanding seemed like a little bit of a stretch from its philosophical source material. If the whole point of the paper is that this framework follows from Kant, especially as the results are weak, I think it's important to make that case rock solid.

**Questions:**

See my questions in 'Weaknesses' above.

Note for authors and AC: I am passingly familiar with the underlying philosophy, but not enough to evaluate whether this paper's characterization of Kant, or the connection of their method to Kant's arguments, is accurate or satisfactory.

**EDIT:** I know it might be a bit late for this, but I've revised my Soundness score up and my review score to just over the acceptance threshold. The authors clarified a fair bit of stuff in the new draft and the results are a lot stronger. I'm still uncertain about the magnitude of the contribution and the soundness of all of the philosophical connections, hence decreased my confidence to 2.

---

> ### Author Response · Authors · 2023-11-15
> **Thank you for your insightful review! (Part1)**
>
> Thanks for your thoughtful and insightful questions! We described the experimental details, including the prompt method, content, and examples in the supplementary material. Below are answers to your queries:
>
>
> Q1: The paper lacks a lot of important details. a. I'm confused by the descriptions of the P, A, and R steps. Sections 3.2–3.4 just discuss the philosophical side and motivation without saying how the model is actually prompted. That seems to me like the most important thing to communicate in the paper. Please include it. like you did with "Understand".
>
> A1: We provide all prompt details and examples in the appendix. To answer your question specifically, the full prompt:
>
> $\text{You are an assistant operating under a Kantian-inspired multilevel thinking structure.}$
>
> $\text{When receiving a problem, do not solve it immediately. Instead, follow these structured instructions:}$
>
> $\text{First, briefly understand this question in the context of time and space step by step.}$
>
> $\text{Quantity: What entitie/events and their quantitative relationships are related to the question? [Your answer here]}$
>
> $\text{Quality: What intrinsic properties and external constraints of these entities/events are related to the question? [Your answer here]}$
>
> $\text{Relation: What is the relationship between these entities/events? [Your answer here]}$
>
> $\text{Modality: Is possibility/impossibility, inevitable/accidental involved in the entities/events related to the question? [Your answer here]}$
>
> $\text{Let's make a brief plan to solve this question step by step: [Your plan here.]}$
>
> $\text{Now, let's execute the plan step by step: [Your solution here.]}$
>
> $\text{Check your answers and correct possible errors: [Your reflect here.]}$
>
> Q2: Also, please be more specific even in the "Understand" section about how your prompting approach works. Do you prompt it four separate times, once for each question? How do you instruct the model outside of just asking it the question?
>
> A2: For the understand part (including subsequent P, A, R), we only use the zero-shot method to ask all the questions at once in the initial instruction, instead of asking these questions step by step. We do not give any instructions, provide any examples or multiple rounds of interaction other than the prompts we show.
>
>
> Q3: Does it work only on instruction-tuned or RLHF models, or is it designed to work with pretrained LMs as well? Can you use few-shot examples? Where would they come from?
>
> A3: In our current experiments, we use RLHF gpt3.5 and gpt4 models. We will continue to explore the performance of UPAR on pre-trained models in the future. In the current experiment, we did not use any few-shot samples. This is to better test whether the LLM can really think according to an epistemological framework similar to humans (including understanding of the core categories proposed by Kant), rather than imitating human handcrafting samples. However, we believe that human handcrafted few-shot samples should further improve the effectiveness of UPAR, as shown in most prompt techniques.
>
> Q4: Is there some set of tasks on which it doesn't work? What changes besides the simplification in UPAR-S might be necessary to make it work in other cases?
>
> A4: In our experiments, the main problem encountered is that the existing LLM benchmarks seem to be too simple: most public data sets do not involve complex category interactions and changes, like what humans face in complex reasoning problems in the real world. In this case, instructing the model to discuss categorical concepts that do not exist in the problem is meaningless and unnecessary. It is based on this that we proposed UPARs to allow the model to decide its own understanding method on simple problems.

---

> ### Author Response · Authors · 2023-11-15
> **(Part2)**
>
> Q5: The results are not very promising. It does yield improvements over zero-shot CoT with GPT-4, but only very small ones, and it's unclear whether they are statistically significant (how big are the test sets? What's the total n being tested on?). The only case of a large gap with GPT-4 zero-shot was on a subset of GSM8k filtered to examples which zero-shot CoT with GPT-4 got wrong — not a fair comparison.
>
> A5:
> We added strong results on SCIBENCH dataset (see general response). Experiment results on complex reasoning dataset including SCIBENCH, GSM8K-Hard and Causal-Judgement show that UPAR method can produce significantly more accurate results better results than existing prompting methods. For the testset size, we use SCIBENCH(695), GSM8K (1319), AQuA (254), CSQA (1221), StrategQA (2290), Causal-Judgement (187), GSM8K-Hard (48).
>
> The selection criterion of GSM8K-Hard is that the GPT4+COT method cannot answer correctly. Additionally, we manually cleared those questions with incorrect answers. The reason why we did this is because most of the problems in GSM8K are relatively simple and do not involve complex category relationships, such as:"A robe takes 2 bolts of blue fiber and half that much white fiber.  How many bolts in total does it take?" In GSM8k-hard, the complexity of the problem has increased significantly, which is why the linear reasoning process of COT cannot work well, such as: "Lorraine and Colleen are trading stickers for buttons. Each large sticker is worth a large button or three small buttons. A small sticker is worth one small button. A large button is worth three small stickers. Lorraine starts with 30 small stickers and 40 large stickers. She trades 90\% of her small stickers for large buttons. She trades 50\% of her large stickers for large buttons and trades the rest of them for small buttons. How many buttons does she have by the end?".  In these questions, step-by-step answers often lead to logical confusion. In the UPAR framework, by allowing  the LLM to automatically analyze basic cognitive concepts in complex contexts, a more comprehensive context can be provided for the LLM to perform subsequent reasoning.
>
> Q6: It's not totally clear to me how deep the relationship goes between the prompting approach and the philosophical backdrop. Especially given the lack of detail in the paper, one can imagine many possible ways of implementing the same idea. Why this way in particular? For example, the assignment of questions to the pure categories of understanding seemed like a little bit of a stretch from its philosophical source material. If the whole point of the paper is that this framework follows from Kant, especially as the results are weak, I think it's important to make that case rock solid.
>
> A6: We provide comprehensive experimental details in the supplementary materials for your reference. Regarding the connection with philosophical method, we admit that we do not fully follow the full implications of Kant's transcendental philosophy. The biggest difference is that in Kant's philosophy, cognitive categories are inherent cognitive structures of humans, even without exposure to real-world data. However, Transformer, the infrastructure of large language models, obviously does not naturally have these capabilities. To alleviate this contradiction, we propose to explicitly extract these cognitive categories from the trained large-scale language model itself and use them as problem-solving context to enhance the reliability of the inference process.
>
> More importantly, we wish to point out that we are trying to bring Kant's transcendental philosophy into current discussions in the field of large language models, rather than claiming that we provide the only way to achieve it. The core question we hope to answer is whether such cognitive structures can provide performance beyond simple prompting techniques. We particularly look forward to follow-up work exploring this direction, particularly discussing the closer integration of these cognitive categories as a "pre"-training/inherent ability with neural networks.
>
> We'd be glad to discuss any unclear details further.

---

> > ### Author Response · Authors · 2023-11-19
> > **Additional Data and details**
> >
> > Dear reviewers, please review our general response for additional strong experimental data and details.

---

### Official Review · Reviewer_Yqz2 · 2023-11-09

**Soundness:** 3 good
**Presentation:** 4 excellent
**Contribution:** 3 good
**Rating:** 6
**Confidence:** 2

**Summary:**

This paper proposes a new prompting framework based on Kant's philosophy to enhance large language models.  It tries to emulate the structure of human cognition within the LLMs. The framework of UPAR: understand, plan, act and reflect, tries to structure the prompt with these four reasoning components based on Kant’s philosophy and even more fine-grained elements of understanding such as time, space,  events and their relationships and more. They show that asking the language model to adhere to these steps of reasoning improves accuracy in question answering/reasoning. They were able to improve GPT-4 results on two benchmarks on causal judgement and grade school math problems (GSM8k) compared to COT prompting.

**Strengths:**

-The paper is very well-written.
-The interdisciplinary aspect of paper is novel and interesting as it applies Kant's philosophy to reasoning structure of LLMs.
-The background information and overview of the related work was done very good and neatly.
-The results show some improvements in reasoning over text compared to baselines.

**Weaknesses:**

--The experimental results are not very strong. GPT3.5 does not show any improvements. GPT-4 has a mixture of results, mostly improves a bit though.

--The fact that they needed to simplify the prompt steps to obtain better results weakens the idea of applicability of the theory in this context. Specially, there are several results that show dividing the input to parts and having step-by-step reasoning is helpful, so I am not sure if Kant's theory is specifically helpful here or dividing the problem to sub-problems in anyways can be helpful. The results are only compared to COT not any other newer variations of step by step reasoning compared here. More baseline might show the advantage of this theory better (?).

--It was not clear how they provided the information about each step of reasoning to the LLM, I could not see additional descriptions in the prompt other than the keywords like understand, plan, etc.

**Questions:**

-How many examples did you provide in the context [input of the LLM]?
-Did you only use the keywords of understand, plan, etc along with an example for in-context learning? and without any further explanation?
-Did you do this step by step? i.e. the output of first step will be the input to the next step? do you concat the output each time to the older input?
More details about the exact interactions with the LLM for obtaining the answer will be helpful.

---

> ### Author Response · Authors · 2023-11-15
> **Thank you for your insightful review!**
>
> Thanks for your thoughtful and insightful questions! We described the experimental details, including the prompt method, content, and examples in the supplementary material. Below are answers to your queries:
>
> Q1: I am not sure if Kant's theory is specifically helpful here or dividing the problem to sub-problems in anyways can be helpful.
>
> A1: For the scalability of our method, we do not manually split any subproblems, nor provide any few-shot samples to instruct the model to do so. Specifically, all understanding, planning, execution, and reflection are completed within a single output of the model. This means that the model decides how to solve problems itself, rather than manually designed by us.
>
> Q2: How many examples did you provide in the context [input of the LLM]?
>
> A2: In order to eliminate the influence of constructing few-shot samples, all our experiments use the zero-shot method. We do not provide any concrete examples of UPAR other than telling the LLM to follow the UPAR framework's prompts. This is also to verify to a certain extent whether the large language model has human-like systematic thinking capabilities, rather than simply imitating samples.
>
> Q3: Did you only use the keywords of understand, plan, etc along with an example for in-context learning? and without any further explanation? -Did you do this step by step? i.e. the output of first step will be the input to the next step?
>
> A3: We only use UPAR related keywords without using any examples or any additional explanation. We do not instruct the model to answer the question step by step. Instead, we use a short prompt at the beginning of the question to guide the model to think in the UPAR way, and the model will generate remaining answers as a single output step.
>
> Q4: Do you concat the output each time to the older input? More details about the exact interactions with the LLM for obtaining the answer will be helpful.
>
> A4: No, except providing the initial prompt, in the experiment we did not perform any manual operations such as connecting or providing new input, correction, etc. Comprehensive details can be found in the supplementary material.
>
> We'd be glad to discuss any unclear details further.
>
> Q5: The experimental results are not very strong. GPT3.5 does not show any improvements. GPT-4 has a mixture of results, mostly improves a bit though.
> A5:  We added strong results on SCIBENCH dataset (see general response). Experiment results on complex reasoning dataset including SCIBENCH, GSM8K-Hard and Causal-Judgement show that UPAR method can produce significantly more accurate results better results than existing prompting methods

---

> > ### Author Response · Authors · 2023-11-19
> > **Additional Data and details**
> >
> > Dear reviewers, please review our general response for additional strong experimental data and details.

---

> > > ### Comment · Reviewer_Yqz2 · 2023-11-22
> > >
> > > Thank you for the response and the additional experiments.  The results show strong improvements in these new scientific questions. However, now that you clarified you do not do any prompt engineering other than prompting with those keywords in the theory, it is very hard to understand why this is effective. And why mostly in a scientific context?

---

> ### Author Response · Authors · 2023-11-22
> **Thank you for your response!**
>
> Thanks for your reply, let us explain further.
>
> 1.Our full prompts as a single system level input.
>
> $\text{You are an assistant operating under a Kantian-inspired multilevel thinking structure.}$
>
> $\text{When receiving a problem, do not solve it immediately. Instead, follow these structured instructions:}$
>
> $\text{First, briefly understand this question in the context of time and space step by step.}$
>
> $\text{Quantity: What entitie/events and their quantitative relationships are related to the question? [Your answer here]}$
>
> $\text{Quality: What intrinsic properties and external constraints of these entities/events are related to the question? [Your answer here]}$
>
> $\text{Relation: What is the relationship between these entities/events? [Your answer here]}$
>
> $\text{Modality: Is possibility/impossibility, inevitable/accidental involved in the entities/events related to the question? [Your answer here]}$
>
> $\text{Let's make a brief plan to solve this question step by step: [Your plan here.]}$
>
> $\text{Now, let's execute the plan step by step: [Your solution here.]}$
>
> $\text{Check your answers and correct possible errors: [Your reflect here.]}$
>
>
> 2.Why UPAR is effective?
>
> As a master of philosophy, Kant devoted himself to answering such a core question in his work Critique of Pure Reason: How do people understand the world and obtain knowledge? He analyzed in detail the multiple levels of human cognitive structure: sensibility, understanding, and reason, and discussed how this cognitive structure gradually elevates massively changing perceptual experiences to true knowledge of the world. Our approach is inspired by Kant's philosophy. Given that large language models are fully learn from the massive data created by humans, we ask: (1). Can large language models operate with similar cognitive structures? (2). Will such a cognitive structure further enhance LLMs capabilities?.
>
> Based on this, we designed the UPAR framework and let LLMs not start solving directly when answering questions, but to generate a thinking process in such a human-like multi-level cognitive structure. In this sense, the core contributions of UPAR are: (1) We introduce human epistemological philosophy as the theoretical foundation, rather than scattered intuitive techniques. This provides a new perspective on developments in the field of prompt engineering and offers the possibility to integrate these techniques. (2) For the purpose of scalability across different tasks just like human, our approach strives to build a thinking framework, rather than manually setting up algorithms-likes prompt step to solve the specific problem. Based on this motivation, we do not use any few-shot samples and do not introduce any additional iterations. In this sense, we have a similar orthogonal relationship with various existing prompt methods.
>
>
> 3.Why mostly in a scientific context?
>
> We believe that UPAR is effective on university-level scientific problems because such problems are sufficiently challenging. We need to point out here the difficulty we encountered during our experiments: finding a suitable public dataset to evaluate LLMs as powerful as GPT4 is difficult. We have found two problems in most of the current benchmarks: 1. The questions are too simple and do not require complex thinking processes. 2. There are a lot of wrong manual answers/unclear questions. Taking GSM8K as an example, a large number of questions are similar to: "A robe takes 2 bolts of blue fiber and half that much white fiber. How many bolts in total does it take?" This type of question itself is difficult to fully demonstrate UPAR effectiveness. Also according to our manual inspection, the proportion of incorrect answers accounts for about 3% in GSM8K. This is also mutually confirmed with the result of GPT4 using Code Interpreter reaching 97% on GSM8K in [1]. It is for the above reasons that we adopted the newer SCIBENCH data set, the more complex causal judgment data set, and manually created the difficult GSM8K-Hard data set. Experiments also show that the UPAR method has significant improvements in these difficult tasks.
>
> Happy Thanksgiving!
>
> Reference: [1]Zhou, Aojun, et al. "Solving challenging math word problems using gpt-4 code interpreter with code-based self-verification." arXiv preprint arXiv:2308.07921 (2023).

---

### Author Response · Authors · 2023-11-19
**General responses to reviewers (part1)**

We are grateful to the reviewers for their insightful and constructive suggestions. Based on this, we improved our work and revised the submitted version of the paper, which is summarized as follows:

1.Addtional strong results

We added strong results on the SCIBENCH dataset,  a recent challenging dataset containing 695 collegiate-level mathematics, chemistry, and physics scientific problems including  covering Fundamental Physics ($\text{fund}$), Thermodynamics ($\text{thermo}$), Classical Mechanics ($\text{class}$), Quantum Chemistry ($\text{quan, chemmc}$), Physical Chemistry ($\text{atkins, matter}$), Calculus ($\text{calc}$), Statistics ($\text{stat}$), and Differential Equations ($\text{diff}$).  Results of SCIBENCH demonstrate that UPAR shows consistent superiority and stability over existing prompting methods baseline on complex reasoning task, even without relying on any few-shot examples or using external tools. For such scientific problems that require complex reasoning processes, we argue that the a priori category context and multi-level cognitive framework in UPAR, instead of linear thinking process, can significantly increase the reasoning capabilities of LLMs.


Below table shows GPT4 model accuracy (\%) on SCIBENCH textbook dataset. We take baseline methods and data from (Wang et al., 2023a): zero-shot learning without the system prompt ($\text{Zero$-$S}$), zero-shot learning with the system prompt ($\text{Zero}$), few-shot learning ($\text{Few}$), CoT prompting under zero-shot ($\text{Zero$+$CoT}$) and few-shot learning ($\text{Few$+$CoT}$) scenarios, few-shot learning that prompts to use Python  ($\text{Few$+$Py}$), and Wolfram Language ($\text{Few$+$Wol}$) as external tools.  The best performing score is highlighted in $\textbf{bold}$ and second-best is $\underline{underlined}$.
$$
\\begin{array}{|c|c|c|c|c|c|c|c|c|c|c|c|}
\\hline
\\text{Setting} & \\text{atkins} & \\text{chemmc} & \\text{quan} & \\text{matter} & \\text{fund} & \\text{class} & \\text{thermo} & \\text{diff} & \\text{stat} & \\text{calc} & \\text{Avg.} \\\\
\\hline
Zero-S & 14.95 & 25.64 & 8.82 & 18.37 & 21.92 & 12.77 & 7.46 & 8.00 & 28.00 & 19.05 & 16.81 \\\\
Zero & 27.10 & 23.08 & 14.71 & 22.45 & 15.07 & 8.51 & 11.94 & 18.00 & 56.00 & \underline{42.86} & 25.09 \\\\
Zero+CoT & \underline{28.04} & 43.59 & 14.71 & 20.41 & 21.92 & \textbf{19.15} & 17.91 & 22.00 & 50.67 & \underline{42.86} & 28.52 \\\\
\\hline
Few & 15.87 & 30.77 & 17.65 & 12.24 & 26.03 & 12.77 & 5.97 & 8.00 & 49.33 & 33.33 & 21.46 \\\\
Few+CoT & 21.05 & \underline{46.15} & 17.65 & 26.53 & 27.40 & 14.00 & 13.43 & 18.00 & 61.33 & 35.71 & 28.35 \\\\
Few+Py & 21.05 & 41.03 & \textbf{38.24} & \underline{28.57} & \underline{38.36} & 17.02 & \underline{29.85} & \underline{34.00} & \underline{69.33} & \underline{42.86} & \underline{35.80} \\\\
Few+Wol & 3.74 & 0.00 & 17.65 & 26.53 & 27.30 & 17.02 & 17.91 & 32.00 & 7.69 & 14.29 & 15.56 \\\\
\\hline
UPAR & \textbf{55.28} & \textbf{62.50} & \underline{35.71} & \textbf{42.37} & \textbf{66.26} & \underline{18.52} & \textbf{34.52} & \textbf{38.18} & \textbf{70.53} & \textbf{61.54} & \textbf{50.65} \\\\
\\hline
\\end{array}
$$

2.Experimental details

We added experimental details, specifically: We evaluated our method using the GPT4 models; all temperatures were set to 0 and top_p to 1. For UPAR, we provide the UPAR prompts as system-level inputs to the model in a single input (see appendix for details). We test without any external tools, using it as only the one-time zero-shot system-level prompt, which means we do not provide manually constructed few-shot samples; do not perform multiple Q\&A iterations; do not manually divide sub-questions, and only use the entire contents of model's single output as the final answer.

Our full prompts as a single system level input:

$\text{You are an assistant operating under a Kantian-inspired multilevel thinking structure.}$

$\text{When receiving a problem, do not solve it immediately. Instead, follow these structured instructions:}$

$\text{First, briefly understand this question in the context of time and space step by step.}$

$\text{Quantity: What entitie/events and their quantitative relationships are related to the question? [Your answer here]}$

$\text{Quality: What intrinsic properties and external constraints of these entities/events are related to the question? [Your answer here]}$

$\text{Relation: What is the relationship between these entities/events? [Your answer here]}$

$\text{Modality: Is possibility/impossibility, inevitable/accidental involved in the entities/events related to the question? [Your answer here]}$

$\text{Let's make a brief plan to solve this question step by step: [Your plan here.]}$

$\text{Now, let's execute the plan step by step: [Your solution here.]}$

$\text{Check your answers and correct possible errors: [Your reflect here.]}$

---

> ### Author Response · Authors · 2023-11-22
> **General responses to reviewers (part2)**
>
> 3.Our motivation and why UPAR is effective:
>
> As a master of philosophy, Kant devoted himself to answering such a core question in his work Critique of Pure Reason: How do people understand the world and obtain knowledge? He analyzed in detail the multiple levels of human cognitive structure: sensibility, understanding, and reason, and discussed how this cognitive structure gradually elevates massively changing perceptual experiences to true knowledge of the world. Our approach is inspired by Kant's philosophy. Given that large language models are fully learn from the massive data created by humans, we ask: (1). Can large language models operate with similar cognitive structures? (2). Will such a cognitive structure further enhance LLMs capabilities?.
>
> Based on this, we designed the UPAR framework and let LLMs not start solving directly when answering questions, but to generate a thinking process in such a human-like multi-level cognitive structure. In this sense, the core contributions of UPAR are: (1) We introduce human epistemological philosophy as the theoretical foundation, rather than scattered intuitive techniques. This provides a new perspective on developments in the field of prompt engineering and offers the possibility to integrate these techniques. (2) For the purpose of scalability across different tasks just like human, our approach strives to build a thinking framework, rather than manually setting up algorithms-likes prompt step to solve the specific problem. Based on this motivation, we do not use any few-shot samples and do not introduce any additional iterations. In this sense, we have a similar orthogonal relationship with various existing prompt methods.
>
> 4.Why UPAR is more effective on scientific problems?
>
> We believe that UPAR is effective on university-level scientific problems because such problems are sufficiently challenging. We need to point out here the difficulty we encountered during our experiments: finding a suitable public dataset to evaluate LLMs as powerful as GPT4 is difficult. We have found two problems in most of the current benchmarks: 1. The questions are too simple and do not require complex thinking processes. 2. There are a lot of wrong manual answers/unclear questions. Taking GSM8K as an example, a large number of questions are similar to: "A robe takes 2 bolts of blue fiber and half that much white fiber. How many bolts in total does it take?" This type of question itself is difficult to fully demonstrate UPAR effectiveness. Also according to our manual inspection, the proportion of incorrect answers accounts for about 3% in GSM8K. This is also mutually confirmed with the result of GPT4 using Code Interpreter reaching 97% on GSM8K in [2]. It is for the above reasons that we adopted the newer SCIBENCH data set, the more complex causal judgment data set, and manually created the difficult GSM8K-Hard data set. Experiments also show that the UPAR method has significant improvements in these difficult tasks.
>
> 5.Error Analysis:
>
> We added detailed error analysis in UPAR, specifically: (1)Understand: LLMs may misunderstand the ambiguous semantic content and generate a deterministic attitude answer rather than seek clarification. (2)Plan: LLMs may produce illogical plans for solving problems, even under the premise of correctly understanding. (3)Execution: LLMs cannot accurately perform computational problems such as floating-point multiplication, matrix operations, calculus, complex algebraic equations, etc. Without relying on external tools, LLMs often output unreliable results or refuse to calculate. (4)Reflection: LLMs may not be able to effectively correct their own errors without external suggestion, such as modifying the original correct answer into an incorrect one, or being unable to effectively identify it's own erroneous calculations and reasoning processes.
>
> Reference: (1)Wang, Xiaoxuan, et al. "Scibench: Evaluating college-level scientific problem-solving abilities of large language models." arXiv preprint arXiv:2307.10635 (2023). https://nips.cc/virtual/2023/74997
>
> (2)Zhou, Aojun, et al. "Solving challenging math word problems using gpt-4 code interpreter with code-based self-verification." arXiv preprint arXiv:2308.07921 (2023).

---

### Meta-Review · Area_Chair_EAvy · 2023-12-23

**Metareview:**

This paper proposes a prompt framework to augment the performance of LLMs, inspired by Kant’s transcendental philosophy (Kant et al., 1934), which consists of four phases: "Understand", "Plan", "Act", and "Reflect", each corresponding to a specific prompt sentence. The reviewers generally think the connection with Kant's theory is interesting, the method is simple and easy to understand, and the ablation results are informative. However, they also pointed out key weaknesses including the precise relation between Kant's theory and UPAR, how the theory will guarantee higher reliability, etc. The fact that the model is only inspired by and formulated to connect with Kant's theory weakens the novelty claims. In addition, it seems insufficient for the paper to just compare with COT, as there have been more advanced prompting methods including self-refine (which was cited in the paper) that also asks a model to provide feedback on its own input (kind of doing ``reflect'' as well); it'd be great if the paper can also compare with such methods to see the advantage. I think it is great that the paper tries to connect their prompting framework with Kant's theory, but it could be strengthened by convincing the community that the proposed framework is really powerful and can achieve better results than existing ones.

**Justification For Why Not Higher Score:**

Please see the weaknesses summarized above. The technical novelty of the paper does not seem significant enough besides that it connects the proposed prompting framework with Kant's theory, and moreover, it needs to compare with more prompting methods than COT to show its  advantage, such as those that conduct (self-)reflection/feedback in different ways.

**Justification For Why Not Lower Score:**

N/A

---

### Decision · Program_Chairs · 2024-01-16

Reject